# Assessing Quality of Ultrasound Attenuation Coefficient Results for Liver Fat Quantification

**DOI:** 10.3390/diagnostics14192171

**Published:** 2024-09-29

**Authors:** Giovanna Ferraioli, Laura Maiocchi, Richard G. Barr, Davide Roccarina

**Affiliations:** 1Dipartimento di Scienze Clinico-Chirurgiche, Diagnostiche e Pediatriche, University of Pavia, 27100 Pavia, Italy; 2UOC Malattie Infettive, Fondazione IRCCS Policlinico San Matteo, 27100 Pavia, Italy; 3Department of Radiology, Northeastern Ohio Medical University, Rootstown, OH 44272, USA; rgbarr@zoominternet.net; 4Southwoods Imaging, Youngstown, OH 44512, USA; 5SOD Medicina Interna ed Epatologia, Azienda Ospedaliera Universitaria Careggi, 50134 Florence, Italy; 6Sherlock Liver Unit and UCL Institute for Liver and Digestive Health, Royal Free Hospital, London NW3 2QG, UK

**Keywords:** attenuation coefficient, ultrasound, MASLD, fat quantification, PDFF, liver steatosis, accuracy studies, chronic liver disease

## Abstract

Background/Objectives: Algorithms for quantifying liver fat content based on the ultrasound attenuation coefficient (AC) are currently available; however, little is known about whether their accuracy increases by applying quality criteria such as the interquartile range-to-median ratio (IQR/M) or whether the median or average AC value should be used. Methods: AC measurements were performed with the Aplio i800 ultrasound system using the attenuation imaging (ATI) algorithm (Canon Medical Systems, Otawara, Tochigi, Japan). Magnetic resonance imaging proton density fat fraction (MRI-PDFF) was the reference standard. The diagnostic performance of the AC median value of 5 measurements (AC-M) was compared to that of AC average value (AC-A) of 5 or 3 acquisitions and different levels of IQR/M for median values or standard deviation/average (SD/A) for average values were also analyzed. Concordance between AC-5M, AC-5A, and AC3A was evaluated with concordance correlation coefficient (CCC). Results: A total of 182 individuals (94 females; mean age, 51.2y [SD: 15]) were evaluated. A total of 77 (42.3%) individuals had S0 steatosis (MRI-PDFF < 6%), 75 (41.2%) S1 (MRI-PDFF 6–17%), 10 (5.5%) S2 (MRI-PDFF 17.1–22%), and 20 (11%) S3 (MRI-PDFF ≥ 22.1%). Concordance of AC-5A and AC-3A with AC-5M was excellent (CCC: 0.99 and 0.96, respectively). The correlation with MRI-PDFF was almost perfect. Diagnostic accuracy of AC-5M, AC-5A, and AC3A was not significantly affected by different levels of IQR/M or SD/A. Conclusions: The accuracy of AC in quantifying liver fat content was not affected by reducing the number of acquisitions (from five to three), by using the mean instead of the median, or by reducing the IQR/M or SD/A to ≤5%.

## 1. Introduction

It has been reported that metabolic dysfunction-associated steatotic liver disease (MASLD) currently has a prevalence of 37.8% and is the leading cause of chronic liver disease worldwide [1]. The disease may be asymptomatic until it reaches the advanced stage with decompensation. It has been estimated that some 20% of individuals with MASLD will develop metabolic dysfunction-associated steatohepatitis (MASH) [2]. On the other hand, liver steatosis is a dynamic process that is reversible with appropriate intervention, such as diet and lifestyle changes [3]. Moreover, a pharmacological treatment for MASH patients has recently been approved [4].

Due to the significant number of individuals with MASLD, the availability of non-invasive tools for an early diagnosis of the disease is critical. Algorithms based on ultrasound (US) attenuation coefficient (AC) estimation have recently been implemented in several US systems, and promising results for liver fat content quantification have been reported in the literature [5].

AC estimation is often performed together with liver stiffness measurements. For the latter, guidelines have recommended a standardized protocol to obtain reliable measurements [6]. Among the several points suggested by this protocol, the interquartile range-to-median ratio (IQR/M), which assesses the variability between consecutive acquisitions, is the most important quality criterion. In fact, studies have shown that when this criterion is not met, the accuracy of liver stiffness is significantly reduced [7,8,9]. For liver stiffness measurements, guidelines recommend using the median value of five acquisitions [6].

Currently, data on the use of some quality criteria for AC estimation, such as number of acquisitions, use of the median or mean value, use of IQR/M, or a similar criterion, such as the standard deviation/average (S/A), i.e., the coefficient of variation, remain scarce [10,11]. In particular, it is unclear whether the accuracy of AC in quantifying liver fat content is affected by different settings of these quality criteria.

The World Federation for Ultrasound in Medicine and Biology (WFUMB) recently published a guidance for liver fat quantification using US-based algorithms, which includes a standardized protocol for AC measurement [5]. The protocol suggests using the median or average value of three to five acquisitions with an IQR/M ≤ 15%. However, these instructions were mostly based on experts’ experience rather than research data.

The objectives of this study were to evaluate whether there was any difference in the accuracy of AC measurements when (a) the average value was used instead of the median; (b) the number of acquisitions was reduced from five to three; or (c) different levels of IQR/M or S/A were used.

## 2. Materials and Methods

For the purposes of this cross-sectional study, the data of individuals previously enrolled in prospective studies comparing the diagnostic performance of attenuation coefficient imaging with that of controlled attenuation parameter using magnetic resonance imaging proton density fat fraction (MRI-PDFF) as the reference standard [12,13] were pooled as a single observation for statistical analysis.

Baseline characteristics and routine biochemistry were collected for this cohort.

AC measurements were performed with the Aplio i800 US system using the attenuation imaging (ATI) algorithm (Canon Medical Systems, Otawara, Tochigi, Japan). ATI quantifies the AC using a real-time color-coded map (Figure 1).

The length of the measurement box was set at 3 cm with the upper edge at 2 cm from the liver capsule. The acquisitions were obtained in the right lobe of the liver, through intercostal spaces and with the patient lying in the supine position. Five consecutive AC acquisitions were performed by two expert operators (GF and LM).

MRI-PDFF was performed with a 1.5 Tesla system (Magnetom Aera, Siemens Healthineers, Erlangen, Germany) using an 18-channel surface coil in combination with a 32-channel coil. For each exam, a non-contrast, complex-based gradient-echo 3D sequence, that provides whole-liver coverage, was obtained and a single breath-hold sequence with six echoes was performed [12]. The detection of liver steatosis (S > 0), significant steatosis (S > 1), and severe steatosis (S > 2) were defined by MRI-PDFF ≥ 6%, ≥17.1%, and ≥22.1%, respectively [14].

All patients gave written informed consent. The study protocol conformed to the ethical guidelines of the current Declaration of Helsinki and received approval from the Ethics Committee (23 August 2017, P-20170022247).

To assess whether the use of quality measures improves the diagnostic accuracy, the diagnostic performance of AC was evaluated for median of 5 measurements (AC-5M) and for cases with (a) average of 5 measurements (AC-5A), (b) average of the first 3 measurements (AC-3A), (c) IQR/5M ≤ 5%, (d) IQR/5M ≤ 10%, (e) IQR/5M ≤ 15%, (f) IQR/5M > 15%, (g) SD/5A ≤ 5%, (h) SD/5A ≤ 10%, (i) SD/5A ≤ 15%, (j) SD/5A > 15%, (k) SD/3A ≤ 5%, (l) SD/3A ≤ 10%, (m) SD/3A ≤ 15%, (n) SD/3A > 15%. AC-5M was considered as the reference to compare the results with. As a quality criterion for the average of five or three measurements, SD/A ratio was used because the interquartile range is a parameter not related to the average.

### Statistical Analysis

Descriptive statistics were produced for demographic characteristics for this study sample of patients. The Shapiro–Wilk test was used to test the normal distribution of quantitative variables. When quantitative variables were normally distributed, the results were expressed as the mean value and SD, otherwise median and IQR were reported.

Qualitative variables were summarized as counts and percentages. The Student’s *t*-test compared means of normally distributed continuous variables, while the Mann–Whitney U-test was used where continuous variables were non-normally distributed. Qualitative variables were analyzed with the Chi-square test or Fisher’s exact test where appropriate.

Lin’s concordance correlation coefficient (CCC) was used to assess the degree of agreement between median and average values of AC obtained from five measurements as well as between the median of five and the average of three measurements. CCC can be expressed as the product of Pearson’s r (the measure of precision) and the bias-correction factor (Cb, the measure of accuracy) [15]. CCC ranges in values from 0 to +1. Agreement was classified as poor (0.00–0.20), fair (0.21–0.40), moderate (0.41–0.60), good (0.61–0.80), or excellent (0.81–1.00). The agreement between AC measurements was further assessed by the Bland–Altman analysis with 95% limits of agreement (LoA). Differences within mean ± 1.96 SD (LoA) indicated high agreement, allowing methods to be used interchangeably if differences were not clinically significant [16].

Univariate Pearson’s r coefficient was used to test correlations between AC and MRI-PDFF. The correlations were categorized as follows: 0.00 to 0.25, none or slight; 0.25 to 0.50, fair to moderate; 0.50 to 0.75, moderate to good; 0.75 to 1.00, almost perfect [17]. Comparison of correlation coefficients was performed with Fisher’s r to z Statistic [18].

The diagnostic performance of AC for staging liver steatosis compared to PDFF (reference standard) was assessed using receiver operating characteristic (ROC) curves and the area under the ROC (AUROC) curve analysis. The optimal threshold was determined using the Youden index to maximize sensitivity and specificity [19]. Comparisons of the AUROCs were performed using the method described by DeLong et al. for correlated data [20].

Data analysis was performed using Jamovi 2.3.28 (Sydney, Australia), SPSS (version 25, IBM, New York, NY, USA), MedCalc (Software for Windows, Version 14.8.1, Ostend, Belgium), and R version 4.2.2 (Core Team 2022) statistical packages. R is a free software environment for statistical computing and graphics.

## 3. Results

Overall, 182 individuals (94 females and 88 males; mean age, 51.2 years [SD: 15]) were included. The baseline characteristics of this cohort are presented in Table 1.

A total of 77 (42.3%) patients had S0 steatosis (MRI-PDFF < 6%), 75 (41.2%) had S1 steatosis (MRI-PDFF 6–17%), 10 (5.5%) had S2 steatosis (MRI-PDFF 17.1–22%), and 20 (11%) had S3 steatosis (MRI-PDFF ≥ 22.1%).

All measurements were obtained with R^2 ≥ 0.90, i.e., meeting the ATI algorithm’s criterion for good quality measurements.

IQR/5M, SD/5A, and SD/3A were ≤30% for all AC measurements.

IQR/5M was >15% for 14 (7.7%) AC measurements, ≤15% for 168 (92.3%), ≤10% for 135 (74.2%), ≤5% for 44 (24.2%) AC measurements. SD/5A was >15% for none of the AC measurements, ≤15% for 182 (100%), ≤10% for 178 (97.8%), ≤5% for 162 (89%) AC measurements. SD/3A was >15% for 4 (2.2%) AC measurements, ≤15% for 178 (97.8%), ≤10% for 173 (95.1%), ≤5% for 116 (63.7%) AC measurements.

The concordance between AC-5M and AC-5A, and the concordance between AC-5M and AC-3A were both excellent (CCC 0.991, 95%CI 0.989–0.994, Pearson’s r = 0.99, Cb 0.99 and CCC 0.96, 95%CI 0.95–0.97, Pearson’s r = 0.96, Cb 0.99, respectively) (Figure 2), but the concordance between AC-5M and AC-5A was significantly better (z statistic 0.68, *p* < 0.001). The mean of differences between AC-5M and AC-5A was −0.001 (LoA: −0.034 to 0.031) while the mean of differences between AC-5M and AC-3A was −0.002 (LoA: −0.071 to 0.066) (Figure 3).

The univariate analysis showed an almost perfect correlation of AC-5M and AC-5A with MRI-PDFF (r = 0.85, *p* < 0.001 and 0.85, *p* < 0.001, respectively), with not a statistically significant difference between the two AC values (*p* = 0.97). The correlation between AC-3A and MRI-PDFF was still almost perfect (r = 0.78, *p* < 0.001) but slightly significantly lower compared to that of AC-5M (*p* = 0.05) (Figure 4).

The correlation of AC-5M, AC-5A, and AC-3A with MRI-PDFF did not significantly change when different levels of IQR/M or SD/A were used (Table 2).

The AUROCs of AC-5M, AC-5A, and AC-3A were 0.95 (0.92–0.98; *p* < 0.001), 0.95 (0.92–0.98; *p* < 0.001), and 0.94 (0.91–0.97; *p* < 0.001), respectively, for detecting S > 0 steatosis (Figure 5), and 0.91 (0.87–0.96; *p* < 0.001), 0.91 (0.87–0.96; *p* < 0.001), and 0.89 (0.81–0.96; *p* < 0.001), respectively, for detecting S > 1 steatosis (Figure 6).

There was not a statistically significant difference between the AUROC of AC-5M and those of AC-5A and AC-3A for detecting S > 0 (z statistic 0.54, *p* = 0.59 and z statistic 0.62, *p* = 0.54) and S > 1 (z statistic 0.09, *p* = 0.93 and z statistic 1.1, *p* = 0.27) steatosis.

For AC-5M, 22 (12.1%) individuals were misclassified for S > 0, and 42 (23.1%) individuals for S > 1.

The diagnostic performance with misclassified cases of different quality measures for detecting S > 0 and S > 1 is shown in Table 3 and Table 4, respectively.

The diagnostic accuracy of AC-5M, AC-5A, and AC-3A for S > 0 was similar and did not improve by applying any of the IQR/M or SD/A values. For S > 1, the overall number of misclassified cases was lower for AC-3A independently form the application of any SD/A value.

## 4. Discussion

Several algorithms for the quantification of liver fat with ultrasound based on the AC measurement have been developed and are currently available on the market; however, the thresholds for detecting and grading steatosis are different between studies [21,22,23,24,25,26,27,28,29,30,31,32,33,34,35,36,37,38,39,40,41,42,43,44,45,46,47,48,49,50,51,52,53,54], and some confounding factors that may affect the readings have been reported [5]. Among them, there is the depth dependence of the AC measurement: a linear decrease of the AC value with the depth has been observed, therefore a standardized protocol is crucial for obtaining consistent results [55].

The influence of quality measures on the AC accuracy is still poorly understood.

Using the AC algorithm from another vendor (iATT, Fujifilm Healthcare, Japan), it has been shown that the correlation with controlled attenuation parameter, assessed with Pearson’s r, was affected by the IQR/M of the acquisitions, dropping from 0.75 for IQR/M ≤ 15% to 0.60 for IQR/M > 15% [56].

Mirroring the quality criteria recommended for liver stiffness measurements, an IQR/M ≤ 30% has been arbitrarily used in some studies evaluating the accuracy of AC algorithms [12,57,58,59,60].

In our study, we found that the diagnostic accuracy of the AC algorithm was not significantly affected by the IQR/M when this value was ≤30%. However, it should be emphasized that only 14 cases had an AC value with an IQR/M > 15 to ≤30%, therefore the finding is robust only for IQR/M values up to 15% and it validates the suggestion of the WFUMB guidance for liver fat quantification [5].

To the best of our knowledge, this is the first study aimed at evaluating the effect of different levels of IQR/M on the AC accuracy. The results of this study show that it is not necessary to try to achieve an IQR/M lower than that recommended by the WFUMB guidelines for fat quantification, namely ≤15%. For the average of five or three measurements, SD/A was used as a quality criterion, and it was found that there were no cases with SD/A > 15% with the average of five acquisitions, whereas only four cases had SD/A > 15% with the average of three acquisitions. The results were the same as those obtained with different levels of IQR/M.

Regarding the number of acquisitions, a study performed in 56 overweight and obese adults using the ultrasound derived fat fraction (UDFF, Siemens Healthineers, Issaquah, WA, USA) algorithm, which combines the AC with the backscatter coefficient, showed that there was not a significant difference in AUROCs based on the number of UDFF measurements (3 vs. 5) [61]. Another study evaluating the performance of the AC-Canon in 139 patients with MASLD found that mean AC values from 1, 2, 3, 5, and 7 valid acquisitions were not significantly different for any grade of liver steatosis (S0 to S3), and that there were no significant statistical differences between the AC values obtained with different numbers of acquisitions in predicting steatosis grades [10].

In our study, we found that the percentage of misclassified cases for the detection of steatosis (S > 0) with AC was similar when the median or average of five or three acquisitions was used. Interestingly, we observed that misclassification improved in cases with significant steatosis (S > 1) when the average of three acquisitions was used. The interpretation of this result is difficult; however, some hypotheses can be formulated. The average of five acquisitions may emphasize the higher variability in AC measurements for cases with significant steatosis (S > 1) whose accuracy is evaluated by combining S2 and S3 grades (S0–S1 vs. S2–S3), so averaging five measures may lead to worse accuracy than a simpler average of three measures. Another hypothesis is that some AC readings may be affected by noise or physiologic variability in cases of higher degrees of liver steatosis. Averaging of three readings may reduce the effect of extreme values and improve steatosis estimation.

There are some limitations to this study. First, it was not possible to evaluate whether the AC accuracy in diagnosing and grading liver steatosis was affected by values obtained with a high variability in acquisitions, i.e., with an IQR/M ≥ 30%, because all measurements in this study sample showed an IQR/M ≤ 30%. Studies performed with another quantitative parameter, namely liver stiffness, have shown that the accuracy decreases when the IQR/M is above 30%. However, the two metrics, i.e., AC values and liver stiffness measurements, are obtained in a completely different manner, so the results obtained with one of the two parameters cannot be directly applied to the other. Second, the generalizability of our findings may be limited to this specific AC algorithm, and the results may not be applicable to other AC algorithms from different vendors. Third, the impact of quality measures on the accuracy of AC in cases with severe steatosis (S > 2) was not assessed because of the low number of individuals with S3 steatosis (n = 20). Fourth, this series included a relatively small number of individuals with severe liver steatosis (S3), which may have limited the strength of the findings for this group. Fifth, AC measurements were performed by expert operators; hence, the results of this study may not be transferable to the general population.

## 5. Conclusions

In our study population, the accuracy of AC in quantifying liver fat content was not affected by reducing the number of acquisitions (from five to three), using the mean instead of the median, or reducing the IQR/M or SD/A to ≤5%. The results of this research study support the protocol for AC measurement suggested by the WFUMB guidance on liver fat content [5], which was mostly based on experts’ experience at the time it was released. Our results suggest that the risk of misclassification of cases with significant steatosis is reduced when the mean AC value of three acquisitions is used. To confirm these findings, further studies in real-world settings are needed.

## Figures and Tables

**Figure 1 diagnostics-14-02171-f001:**
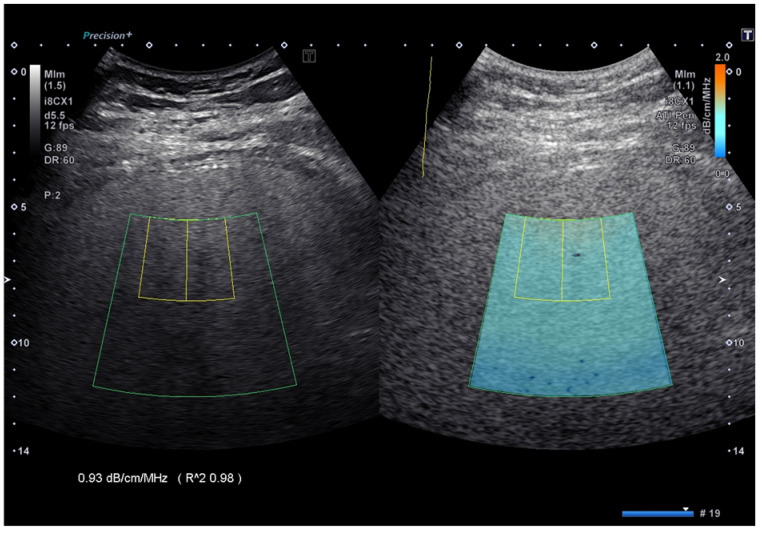
Attenuation coefficient implemented on the Aplio i-series ultrasound system (Canon Medical Systems, Japan). The attenuation coefficient values are color-coded, allowing the visualization of areas of artifacts and to avoid including them in the measurement box. The reliability of the measurement is displayed as an R^2 value, which is a coefficient of determination, and the best quality of the measurement is obtained with an R^2 ≥ 0.90.

**Figure 2 diagnostics-14-02171-f002:**
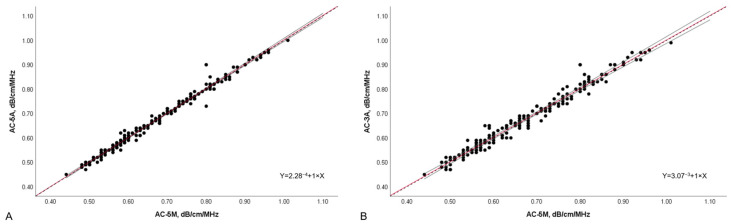
Scatter plot of AC-5A and AC-5M (**A**), and AC-3A and AC-5M (**B**), with linear equation. Dots represent values of attenuation coefficient and the position of each dot on the horizontal and vertical axis indicates values for an individual data point. The continuous red line constitutes the best fit line with the continuous grey lines representing the 95% confidence interval, while the black dashed line indicates the perfect correlation (Y = 0 + 1 × X). AC-5M, attenuation coefficient (median of five measurements); AC-5A, attenuation coefficient (average of five measurements); AC-3A, attenuation coefficient (average of 3 measurements); dB/cm/MHz, decibel/centimeter/megahertz.

**Figure 3 diagnostics-14-02171-f003:**
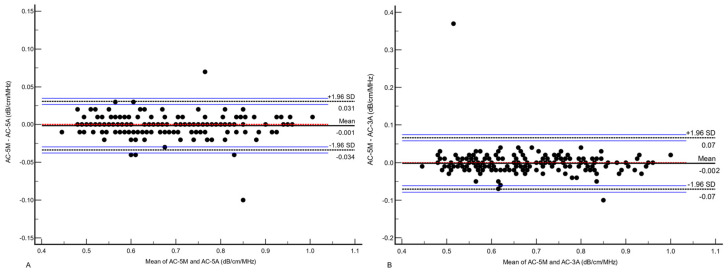
Bland–Altmann plot of the differences between AC-5M and AC-5A (**A**), and AC-5M and AC-3A (**B**) values. The dots represent the difference between paired AC-M and AC-A measurements on the horizontal axis against the average of the paired AC-M and AC-A measurements on the vertical axis. The continuous black line represents the mean of differences; the dotted red line represents the zero line which indicates that for every point on this line the two methods give identical results; the dashed black lines define the 95% limits of agreement with their 95% confidence interval represented by the blue lines. AC-5M, attenuation coefficient (median of five measurements); AC-5A, attenuation coefficient (average of five measurements); AC-3A, attenuation coefficients (average of three measurements); dB/cm/MHz, decibel/centimeter/megahertz; SD, standard deviation.

**Figure 4 diagnostics-14-02171-f004:**
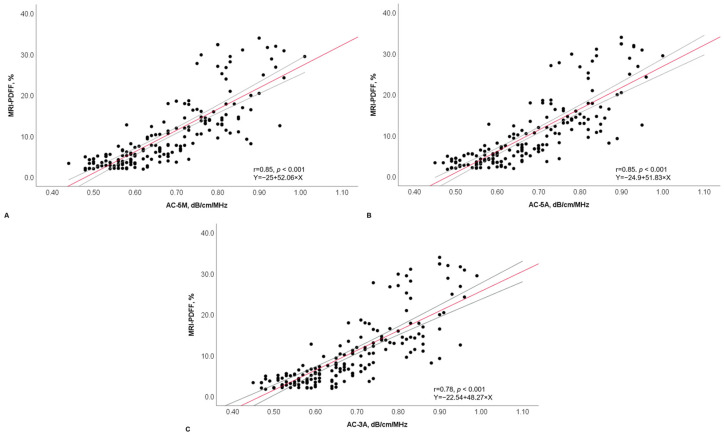
Scatter plot of AC-5M (**A**), AC-5A (**B**), and AC-3A (**C**) with MRI-PDFF, with r and linear equation. Dots represent values of attenuation coefficient and MRI-PDFF values and the position of each dot on the horizontal and vertical axis indicates values for an individual data point. The continuous red line constitutes the best fit line with the continuous grey lines representing the 95% confidence interval. AC-5M, attenuation coefficient (median of five measurements); AC-5A, attenuation coefficient (average of five measurements); AC-3A, attenuation coefficients (average of three measurements); dB/cm/MHz, decibel/centimeter/megahertz; MRI-PDFF, magnetic resonance imaging proton density fatty fraction; r, correlation coefficient.

**Figure 5 diagnostics-14-02171-f005:**
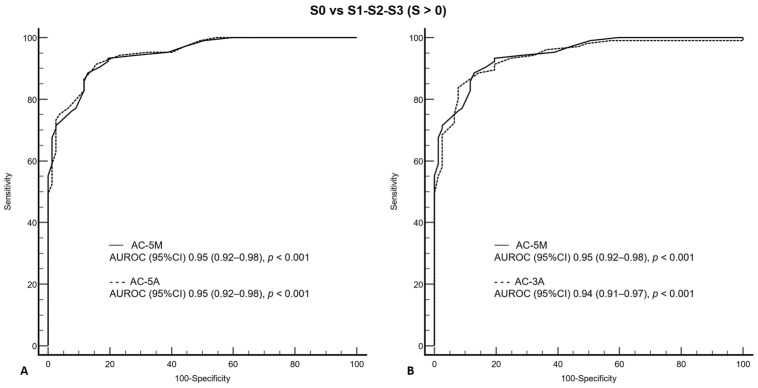
Comparison of receiver operating characteristic curves for AC-5M and AC-5A (**A**), and for AC-5M and AC-3A (**B**) for S0 vs. S1–S3, as defined by MRI-PDFF ≥ 6%. AC-5M, attenuation coefficient (median of five measurements); AC-5A, attenuation coefficient (average of five measurements); AC-3A, attenuation coefficient (average of three measurements); AUROC, area under the receiver operating characteristic (curve); CI, confidence interval; MRI-PDFF, magnetic resonance imaging proton density fatty fraction; *p*, probability of α type I error.

**Figure 6 diagnostics-14-02171-f006:**
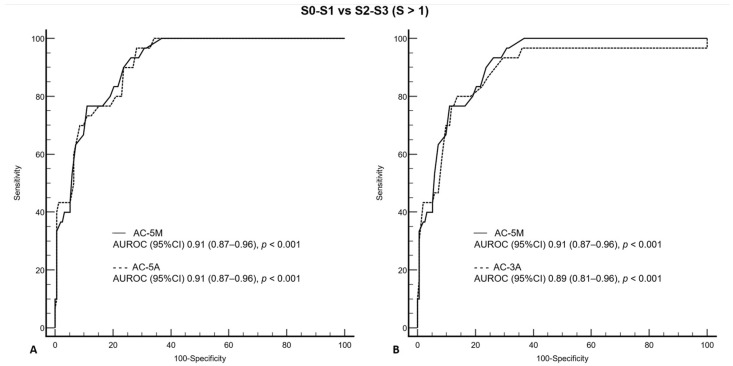
Comparison of receiver operating characteristic curves for AC-5M and AC-5A (**A**), and for AC-5M and AC-3A (**B**) for S0–S1 vs. S2–S3, as defined by MRI-PDFF ≥ 17.1%. AC-5M, attenuation coefficient,(median of five measurements); AC-5A, attenuation coefficient (average of five measurements); AC-3A, attenuation coefficient (average of three measurements); AUROC, area under the receiver operating characteristic (curve); CI, confidence interval; MRI-PDFF, magnetic resonance imaging proton density fatty fraction; *p*, probability of α type I error.

**Table 1 diagnostics-14-02171-t001:** Baseline characteristics of the study cohort.

Variables	Overall (n = 182)	MR-PDFF < 6% (n = 77)	MR-PDFF ≥ 6% (n = 105)	*p*
Age, y	51.16 ± 15	47.2 ± 15.7	54.1 ± 13	0.002
Female, n (%)	94 (51.6)	51 (66.2)	43 (41)	0.001
BMI, kg/m^2^	29.8 ± 4.9	28.3 ± 4	30.9 ± 5.3	<0.001
Waist circumference	102.9 ± 12.4	97.2 ± 11.9	107.1 ± 11	<0.001
Diabetes, n (%)	30 (16.7)	6 (8)	24 (22.9)	0.009
AST, IU/L	21 (13)	18 (7)	24 (16)	<0.001
ALT, IU/L	25 (20)	24 (16)	32 (29)	<0.001
Glycemia, mg/dL	97 ± 18.6	89.6 (10.4)	103.2 (21.5)	<0.001
Triglycerides, mg/dL	103 (69)	87.5 (39)	123 (97)	<0.001
Cholesterol, mg/dL	192.6 ± 46.3	195.7 ± 46.9	190.1 ± 45.9	0.48
Platelet, 109/L	244 ± 65.4	251 ± 66.6	239.2 ± 65.1	0.27
GGT, IU/L	27 (27)	16 (14)	36 (33)	<0.001
MRI-PDFF, %	7.2 (9.9)	3.7 (2.3)	13 (9.8)	<0.001
AC-5M, dB/cm/MHz	0.68 ± 0.13	0.57 ± 0.06	0.76 ± 0.10	<0.001
AC-5A, dB/cm/MHz	0.68 ± 0.13	0.57 ± 0.06	0.76 ± 0.10	<0.001
AC-3A, dB/cm/MHz	0.68 ± 0.13	0.57 ± 0.06	0.76 ± 0.11	<0.001

Numbers in parentheses represent interquartile range unless otherwise specified. Abbreviations: AC-5M, attenuation coefficient (median of five measurements); AC-5A, attenuation coefficient (average of five measurements); AC-3A, attenuation coefficient (average of three measurements); AST, aspartate transaminase; ALT, alanine transaminase; BMI, body mass index; GGT, gamma-glutamyl transferase; MRI-PDFF, magnetic resonance imaging proton density fat fraction; *p*, probability of α type I error.

**Table 2 diagnostics-14-02171-t002:** Correlation coefficients of AC-5M, AC-5A, and AC-3A with MRI-PDFF after the application of different quality criteria.

Observations	r Coefficient	*p* *	z Statistic	*p* **
AC-5M: 182	0.85	<0.001		
IQR/5M ≤ 5%: 44	0.87	<0.001	−0.33	0.74
IQR5/M ≤ 10%: 135	0.85	<0.001	0.13	0.90
IQR/5M ≤ 15%: 168	0.85	<0.001	0.17	0.87
IQR/5M > 15%: 14	0.91	<0.001	0.87	0.38
AC-5A: 182	0.86	<0.001	0.35	0.72
SD/5A ≤ 5%: 162	0.86	<0.001	0.34	0.73
SD/5A ≤ 10%: 178	0.86	<0.001	0.35	0.73
SD/5A ≤ 15%: 182	0.86	<0.001	0.35	0.73
AC-3A: 182	0.78	<0.001	−2.00	0.05
SD/3A ≤ 5%: 116	0.82	<0.001	−0.83	0.41
SD/3A ≤ 10%: 173	0.82	<0.001	−0.93	0.35
SD/3A ≤ 15%: 178	0.78	<0.001	−1.98	0.05
SD/3A > 15%: 4	0.96	<0.001	0.69	0.49

* Comparison with correlation coefficient of AC-5M. AC-5M, attenuation coefficient (median of five measurements); AC-5A, attenuation coefficient (average of five measurements); AC-3A, attenuation coefficient (average of three measurements); IQR, interquartile range; MRI-PDFF, magnetic resonance imaging proton density fat fraction; *p* *, probability of α type I error for correlation coefficient; *p* **, probability of α type I error for correlation coefficients comparison; r coefficient, Pearson’s correlation coefficient; SD, standard deviation; z statistic, Fisher’s to z statistic for correlation coefficient comparison.

**Table 3 diagnostics-14-02171-t003:** Performance with misclassified cases of AC-5M, AC-5A, and AC-3A for detecting S > 0 liver steatosis as defined by MRI-PDFF ≥ 6%.

Parameter	Algorithm (Observations n)	S0 vs. S1–S3
AUROC (95% CI)	AC-5M (overall) ^†^ IQR/5M ≤ 5% (44) IQR/5M ≤ 10% (135) IQR/5M ≤ 15% (168) IQR/5M > 15% (14) AC-5A (overall) ^†^ SD/5A ≤ 5% (162) SD/5A ≤ 10% (178) SD/5A ≤ 15% (182) SD/5A > 15% (0) AC-3A (overall) ^†^ SD/3A ≤ 5% (116) SD/3A ≤ 10% (173) SD/3A ≤ 15% (178) SD/3A > 15% (4)	0.95 (0.92–0.98) 0.97 (0.93–1.00) *p* = 0.33 * 0.93 (0.89–0.97) *p* = 0.51 * 0.94 (0.91–0.97) *p* = 0.89 * 0.96 (0.85–1.00) *p* = 0.87 * 0.95 (0.92–0.98) *p* = 0.93 * 0.95 (0.92–0.98) *p* = 1.00 * 0.95 (0.92–0.98) *p* = 1.00 * 0.95 (0.92–0.98) *p* = 0.93 * ---- 0.94 (0.91–0.97) *p* = 0.66 * 0.92 (0.88–0.97) *p* = 0.29 * 0.95 (0.92–0.98) *p* = 1.00 * 0.94 (0.90–0.97) *p* = 0.67 * 1.00 (1.00–1.00) *p* < 0.001 *
Sensitivity % (95% CI)	AC-5M (overall) ^†^ IQR/5M ≤ 5% IQR/5M ≤ 10% IQR/5M ≤ 15% IQR/5M > 15% ^ AC-5A (overall) ^†^ SD/5A ≤ 5% SD/5A ≤ 10% SD/5A ≤ 15% SD/5A > 15% AC-3A (overall) ^†^ SD/3A ≤ 5% SD/3A ≤ 10% SD/3A ≤ 15% SD/3A > 15% ^§^	88.6 (80.9–94.0) 84.9 (68.1–94.9) 88.6 (80.1–94.4) 89.1 (81.4–94.4) 75.0 (19.4–99.4) 91.4 (84.4–96.0) 91.4 (83.8–96.2) 91.2 (83.9–95.9) 91.4 (84.4–96.0) ---- 83.8 (75.4–90.3) 84.6 (74.7–91.8) 84.8 (76.2–91.3) 83.5 (74.9–90.1) 100 (15.8–100)
Specificity % (95% CI)	AC-5M (overall) ^†^ IQR/5M ≤ 5% IQR/5M ≤ 10% IQR/5M ≤ 15% IQR/5M > 15% ^ AC-5A (overall) ^†^ SD/5A ≤ 5% SD/5A ≤ 10% SD/5A ≤ 15% SD/5A > 15% AC-3A (overall) ^†^ SD/3A ≤ 5% SD/3A ≤ 10% SD/3A ≤ 15% SD/3A > 15% ^§^	87.0 (77.4–93.6) 90.9 (58.7–99.8) 80.9 (66.7–90.9) 86.6 (76.0–93.7) 90.0 (55.5–99.8) 84.4 (74.4–91.7) 84.1 (73.3–91.8) 84.2 (74.0–91.6) 84.4 (74.4–91.7) ---- 92.2 (83.8–97.1) 84.2 (68.8–94.0) 91.9 (83.2–97.0) 92.0 (83.4–97.0) 100 (15.8–100)
PPV % (95% CI)	AC-5M (overall) ^†^ IQR/5M ≤ 5% IQR/5M ≤ 10% IQR/5M ≤ 15% IQR/5M > 15% ^ AC-5A (overall) ^†^ SD/5A ≤ 5% SD/5A ≤ 10% SD/5A ≤ 15% SD/5A > 15% AC-3A (overall) ^†^ SD/3A ≤ 5% SD/3A ≤ 10% SD/3A ≤ 15% SD/3A > 15% ^§^	90.2 (82.9–95.3) 96.6 (88.2–99.9) 89.7 (81.3–95.2) 90.9 (83.4–95.8) 75.0 (30.1–95.4) 88.9 (81.4–94.1) 88.5 (80.4–94.1) 88.6 (80.9–94.0) 88.9 (81.4–94.1) ---- 93.6 (87.1–97) 91.7 (84.0–95.8) 93.3 (86.6–96.8) 93.5 (86.9–96.9) 100 (15.8–100)
NPV % (95% CI)	AC-5M (overall) ^†^ IQR/5M ≤ 5% IQR/5M ≤ 10% IQR/5M ≤ 15% IQR/5M > 15% ^ AC-5A (overall) ^†^ SD/5A ≤ 5% SD/5A ≤ 10% SD/5A ≤ 15% SD/5A > 15% AC-3A (overall) ^†^ SD/3A ≤ 5% SD/3A ≤ 10% SD/3A ≤ 15% SD/3A > 15% ^§^	84.8 (75.0–91.9) 66.7 (38.4–88.2) 79.2 (65.0–89.5) 84.1 (73.3–91.8) 90 (62–98) 87.8 (78.2–94.3) 87.9 (77.5–94.6) 87.7 (77.9–94.2) 87.8 (78.2–94.3) ---- 80.7 (72.9–86.6) 72.7 (60.9–82) 81.9 (73.9–87.9) 80.2 (72.3–86.3) 100 (15.8–100)
LR+ (95% CI)	AC-5M (overall) ^†^ IQR/5M ≤ 5% IQR/5M ≤ 10% IQR/5M ≤ 15% IQR/5M > 15% ^ AC-5A (overall) ^†^ SD/5A ≤ 5% SD/5A ≤ 10% SD/5A ≤ 15% SD/5A > 15% AC-3A (overall) ^†^ SD/3A ≤ 5% SD/3A ≤ 10% SD/3A ≤ 15% SD/3A > 15% ^§^	6.82 (3.81–12.21) 9.33 (1.43–60.82) 4.63 (2.56–8.37) 6.63 (3.60–12.23) 7.5 (1.07–52.38) 5.87 (3.48–9.90) 5.73 (3.32–9.89) 5.77 (3.42–9.74) 5.87 (3.48–9.90) ---- 10.76 (4.97–23.3) 5.36 (2.56–11.24) 10.46 (4.84–22.64) 10.44 (4.82–22.59) ----
LR− (95% CI)	AC-5M (overall) ^†^ IQR/5M ≤ 5% IQR/5M ≤ 10% IQR/5M ≤ 15% IQR/5M > 15% ^ AC-5A (overall) ^†^ SD/5A ≤ 5% SD/5A ≤ 10% SD/5A ≤ 15% SD/5A > 15% AC-3A (overall) ^†^ SD/3A ≤ 5% SD/3A ≤ 10% SD/3A ≤ 15% SD/3A > 15% ^§^	0.13 (0.08–0.23) 0.17 (0.07–0.38) 0.14 (0.08–0.26) 0.13 (0.07–0.22) 0.28 (0.05–1.54) 0.10 (0.05–0.19) 0.10 (0.05–0.20) 0.10 (0.06–0.20) 0.10 (0.05–0.19) ---- 0.18 (0.11–0.27) 0.18 (0.11–0.31) 0.16 (0.10–0.26) 0.18 (0.12–0.28) 0
Total misclassified cases, n (%)	AC-5M (overall) ^†^ IQR/5M ≤ 5% IQR/5M ≤ 10% IQR/5M ≤ 15% IQR/5M > 15% ^ AC-5A (overall) ^†^ SD/5A ≤ 5% SD/5A ≤ 10% SD/5A ≤ 15% SD/5A > 15% AC-3A (overall) ^†^ SD/3A ≤ 5% SD/3A ≤ 10% SD/3A ≤ 15% SD/3A > 15% ^§^	22 (12 FN + 10 FP) (12.1) 6 (5 FN + 1 FP) (13.6) 19 (10 FN + 9 FP) (14.1) 20 (11 FN + 9 FP) (11.9) 2 (1 FN + 1 FP) (14.3) 21 (9 FN + 12 FP) (11.5) 19 (8 FN + 11 FP) (11.7) 21 (9 FN + 12 FP) (11.8) 21 (9 FN + 12 FP) (11.5) ---- 23 (17 FN + 6 FP) (12.6) 18 (12 FN + 6 FP) (15.5) 21 (15 FN + 6 FP) (12.1) 23 (17 FN + 6 FP) (12.9) 0

*, AUROC comparison with AC-5M AUROC; ^†^, IQR/M and SD/A were ≤30% for all AC measurements; ^, only 14 individuals had IQR/M > 15%; ^§^, only 4 individuals had SD/3A > 15%. AC-5M, attenuation coefficient (median of five measurements); AC-5A, attenuation coefficient (average of five measurements); AC-3A, attenuation coefficient (average of three measurements); dB/cm/MHz, decibel/centimeter/megahertz; CI, confidence interval; FN, false negative; FP, false positive; IQR/M, interquartile range to median ratio; SD/A, standard deviation to average ratio; LR+, positive likelihood ratio; LR, negative likelihood ratio; MRI-PDFF, magnetic resonance imaging proton density fat fraction; NPV, negative predictive value; *p*, probability of α type I error; PPV, positive predictive value; S, steatosis grade; SD, standard deviation.

**Table 4 diagnostics-14-02171-t004:** Performance with misclassified cases of AC-5M, AC-5A, and AC-3A for detecting S > 1 liver steatosis as defined by MRI-PDFF ≥ 17.1%.

Parameter	Algorithm (Observations n)	S0–S1 vs. S2–S3
AUROC	AC-5M (overall) ^†^ IQR/5M ≤ 5% (44) IQR/5M ≤ 10% (135) IQR/5M ≤ 15% (168) IQR/5M > 15% (14) AC-5A (overall) ^†^ SD/5A ≤ 5% (162) SD/5A ≤ 10% (178) SD/5A ≤ 15% (182) SD/5A > 15% (0) AC-3A (overall) ^†^ SD/3A ≤ 5% (116) SD/3A ≤ 10% (173) SD/3A ≤ 15% (178) SD/3A > 15% (4)	0.91 (0.87–0.96) 0.89 (0.79–0.98) *p* = 0.65 * 0.90 (0.85–0.95) *p* = 0.72 * 0.91 (0.86–0.96) *p* = 0.86 * 0.96 (0.85–1.00) *p* = 0.42 * 0.91 (0.87–0.96) *p* = 1 * 0.92 (0.87–0.96) *p* = 0.93 * 0.92 (0.87–0.96) *p* = 0.90 * 0.91 (0.87–0.96) *p* = 1 * ---- 0.89 (0.81–0.96) *p* = 0.65 * 0.90 (0.85–0.96) *p* = 0.79 * 0.92 (0.87–0.96) *p* = 0.76 * 0.88 (0.81–0.96) *p* = 0.57 * 1.00 (1.00–1.00) *p* ≤ 0.01 *
Sensitivity %	AC-5M (overall) ^†^ IQR/5M ≤ 5% IQR/5M ≤ 10% IQR/5M ≤ 15% IQR/5M > 15% ^ AC-5A (overall) ^†^ SD/5A ≤ 5% SD/5A ≤ 10% SD/5A ≤ 15% SD/5A > 15% AC-3A (overall) ^†^ SD/3A ≤ 5% SD/3A ≤ 10% SD/3A ≤ 15% SD/3A > 15% ^§^	93.3 (77.9–99.2) 100 (76.8–100) 92.9 (76.5–99.1) 93.1 (77.2–99.2) 100 (2.5–100) 96.7 (82.8–99.9) 96.2 (80.4–99.9) 96.3 (81.0–99.9) 96.7 (82.8–99.9) ---- 80.0 (61.4–92.3) 82.6 (61.2–95.1) 80.8 (60.7–93.5) 78.6 (59.1–91.7) 100 (15.8–100)
Specificity %	AC-5M (overall) ^†^ IQR/5M ≤ 5% IQR/5M ≤ 10% IQR/5M ≤ 15% IQR/5M > 15% ^ AC-5A (overall) ^†^ SD/5A ≤ 5% SD/5A ≤ 10% SD/5A ≤ 15% SD/5A > 15% AC-3A (overall) ^†^ SD/3A ≤ 5% SD/3A ≤ 10% SD/3A ≤ 15% SD/3A > 15% ^§^	73.7 (65.9–80.5) 66.7 (47.2–82.7) 69.2 (59.5–77.7) 71.9 (63.7–79.2) 92.3 (64.0–99.8) 71.7 (63.8–78.7) 71.3 (63.0–78.8) 71.5 (63.6–78.6) 71.7 (63.8–78.7) ---- 86.2 (79.7–91.2) 85.0 (76.0–91.5) 85.7 (79.0–90.9) 86.0 (74.9–91.1) 100 (15.8–100)
PPV %	AC-5M (overall) ^†^ IQR/5M ≤ 5% IQR/5M ≤ 10% IQR/5M ≤ 15% IQR/5M > 15% ^ AC-5A (overall) ^†^ SD/5A ≤ 5% SD/5A ≤ 10% SD/5A ≤ 15% SD/5A > 15% AC-3A (overall) ^†^ SD/3A ≤ 5% SD/3A ≤ 10% SD/3A ≤ 15% SD/3A > 15% ^§^	41.2 (29.4–53.8) 58.3 (36.6–77.9) 44.1 (31.2–57.6) 40.9 (29.0–53.7) 50.0 (13.2–86.8) 40.3 (28.9–52.5) 39.1 (27.1–52.1) 37.7 (26.3–50.2) 40.3 (28.9–52.5) ---- 53.3 (37.9–68.3) 57.6 (44.7–69.5) 50 (39.2%–60.8) 51.2 (40.3–62.0) 100 (15.8–100)
NPV %	AC-5M (overall) ^†^ IQR/5M ≤ 5% IQR/5M ≤ 10% IQR/5M ≤ 15% IQR/5M > 15% ^ AC-5A (overall) ^†^ SD/5A ≤ 5% SD/5A ≤ 10% SD/5A ≤ 15% SD/5A > 15% AC-3A (overall) ^†^ SD/3A ≤ 5% SD/3A ≤ 10% SD/3A ≤ 15% SD/3A > 15% ^§^	98.3 (93.8–99.8) 100 (83.2–100) 97.4 (90.8–99.7) 98.0 (93.1–99.8) 100 (73.5–100) 99.1 (95.0–100) 99.0 (94.5–100) 99.1 (95–100) 99.1 (95–100) ---- 95.6 (90.7–98.4) 95.2 (89–98) 96.2 (92–98.2) 95.6 (91.3–97.8) 100 (15.8–100)
LR+	AC-5M (overall) ^†^ IQR/5M ≤ 5% IQR/5M ≤ 10% IQR/5M ≤ 15% IQR/5M > 15% ^ AC-5A (overall) ^†^ SD/5A ≤ 5% SD/5A ≤ 10% SD/5A ≤ 15% SD/5A > 15% AC-3A (overall) ^†^ SD/3A ≤ 5% SD/3A ≤ 10% SD/3A ≤ 15% SD/3A > 15% ^§^	3.55 (2.67–4.71) 3.00 (1.81–4.98) 3.01 (2.23–4.07) 3.32 (2.50–4.41) 13 (1.98–85.46) 3.42 (2.63–4.44) 3.35 (2.54–4.42) 3.38 (2.60–4.40) 3.42 (2.63–4.44) ---- 5.79 (3.75–8.95) 5.49 (3.27–9.21) 5.65 (3.65–8.76) 5.61 (3.61–8.73) ----
LR−	AC-5M (overall) ^†^ IQR/5M ≤ 5% IQR/5M ≤ 10% IQR/5M ≤1 5% IQR/5M > 15% ^ AC-5A (overall) ^†^ SD/5A ≤ 5% SD/5A ≤ 10% SD/5A ≤ 15% SD/5A > 15% AC-3A (overall) ^†^ SD/3A ≤ 5% SD/3A ≤ 10% SD/3A ≤ 15% SD/3A > 15% ^§^	0.09 (0.02–0.35) 0 0.10 (0.03–0.40) 0.10 (0.03–0.37) 0 0.05 (0.01–0.35) 0.05 (0.01–0.37) 0.05 (0.01–0.36) 0.05 (0.01–0.32) ---- 0.23 (0.11–0.48) 0.20 (0.08–0.50) 5.65 (3.65–8.76) 0.25 (0.12–0.51) 0
Total misclassified cases, n (%)	AC-5M (overall) ^†^ IQR/5M ≤ 5% IQR/5M ≤ 10% IQR/5M ≤ 15% IQR/5M > 15% ^ AC-5A (overall) ^†^ SD/5A ≤ 5% SD/5A ≤ 10% SD/5A ≤ 15% SD/5A > 15% AC-3A (overall) ^†^ SD/3A ≤ 5% SD/3A ≤ 1 0% SD/3A ≤ 15% SD/3A > 15% ^§^	42 (2 FN + 40 FP) (23.1) 10 (0 FN + 10 FP) (22.7) 35 (2 FN + 33 FP) (25.9) 41 (2 FN + 39 FP) (24.4) 1 (0 FN + 1 FP) (14.3) 44 (1 FN + 43 FP) (24.2) 40 (1 FN + 39 FP) (24.7) 44 (1 FN + 43 FP) (24.7) 44 (1 FN + 43 FP) (24.2) ---- 27 (6 FN + 21 FP) (14.8) 18 (4 FN + 14 FP) (15.5) 26 (5 FN + 21 FP) (15.0) 27 (6 FN + 21 FP) (15.2) 0

*, AUROC comparison with AC-5M AUROC; ^†^, IQR/M and SD/A were ≤30% for all AC measurements; ^, only 14 individuals had IQR/M > 15%; ^§^, only 4 individuals had SD/3A > 15%. AC-5M, attenuation coefficient (median of five measurements); AC-5A, attenuation coefficient (average of five measurements); AC-3A, attenuation coefficient (average of three measurements); dB/cm/MHz, decibel/centimeter/megahertz; CI, confidence interval; FN, false negative; FP, false positive; IQR/M, interquartile range to median ratio; SD/A, standard deviation to average ratio; LR+, positive likelihood ratio; LR, negative likelihood ratio; MRI-PDFF, magnetic resonance imaging proton density fat fraction; *p*, probability of α type I error; NPV, negative predictive value; PPV, positive predictive value; S, steatosis grade; SD, standard deviation.

## Data Availability

The data presented in this study are available on request from the corresponding author. The data are not publicly available due to the privacy of the participants.

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
