# Peer review of "Assessing Quality of Ultrasound Attenuation Coefficient Results for Liver Fat Quantification"

_diagnostics, 2024, doi:10.3390/diagnostics14192171_

Round 1

Reviewer 1 Report

Comments and Suggestions for Authors

Ultrasound-based hepatic fat quantification is a popular research area. The efforts to standardize measurement protocols are of great importance. Currently, limited information exists in the literature about the number of required measurements or the optimal cut-off values for IQR/M or SD/A. Therefore, this well-designed and well-written article addresses a clinically relevant issue. I think minor revisions will be appropriate for clarification of some points. 

- Because the authors stated that "overall, 182 patients met inclusion criteria," it would be better to briefly describe the inclusion criteria for clarification.

- It is important to state that all measurements obtained with R^2 ≥ 0.90.

- Because the expert operators performed 5 consecutive measurements, I think it worths mentioning which 3 of 5 measurements were included in analyses of AC-3A.

- If misclassification is improved with the average of 3 acquisitions, is it possible to recommend AC-3A with SD/A less than 15% in real-world practice?

-In my opinion, the first limitation is not necessary because, without a quality criterion, reliability of the dataset could be considered low in previous studies in which the included patients of the current study were pooled. Instead, the generalizability of the findings may be limited to the AC values of the vendor and the results may not be transferable to AC values of other vendors since the algorithms of AC acquisitions were different between vendors.

Author Response

Dear Reviewer,

Ultrasound-based hepatic fat quantification is a popular research area. The efforts to standardize measurement protocols are of great importance. Currently, limited information exists in the literature about the number of required measurements or the optimal cut-off values for IQR/M or SD/A. Therefore, this well-designed and well-written article addresses a clinically relevant issue. I think minor revisions will be appropriate for clarification of some points. 

R: Thank you for your evaluation.

- Because the authors stated that "overall, 182 patients met inclusion criteria," it would be better to briefly describe the inclusion criteria for clarification.

R: Thanks for pointing out that. In the revised manuscript, the sentence has been modified as follows: 182 patients were included.

- It is important to state that all measurements obtained with R^2 ≥ 0.90.

R: In the results section of the revised manuscript the following sentence has been added: All measurements were obtained with R^2 ≥ 0.90, i.e. meeting the ATI algorithm's criterion for good quality measurements.

- Because the expert operators performed 5 consecutive measurements, I think it worths mentioning which 3 of 5 measurements were included in analyses of AC-3A.

R: The first three measurements. It has been specified in the M&M section of the revised manuscript.

- If misclassification is improved with the average of 3 acquisitions, is it possible to recommend AC-3A with SD/A less than 15% in real-world practice?

R: This is an interesting finding, but we believe that further studies in larger series are needed before it can be recommended in the real-word practice. 

-In my opinion, the first limitation is not necessary because, without a quality criterion, reliability of the dataset could be considered low in previous studies in which the included patients of the current study were pooled. Instead, the generalizability of the findings may be limited to the AC values of the vendor and the results may not be transferable to AC values of other vendors since the algorithms of AC acquisitions were different between vendors.

R: In our series there were very few cases with an IQR/M ≥30%, therefore we couldn’t assess whether there is a decrease in accuracy when the IQR/M is above that range. We agree that the findings of our study are limited to the ATI algorithm, and we have added the following statement in the limitation section of the revised manuscript: The generalizability of our findings may be limited to this specific AC algorithm, and the results may not be applicable to other AC algorithms from different vendors.

Reviewer 2 Report

Comments and Suggestions for Authors

General Comments: The manuscript titled "Assessing Quality of Ultrasound Attenuation Coefficient Results for Liver Fat Quantification" provides a valuable contribution to the field of liver fat quantification, offering a detailed evaluation of ultrasound-based attenuation coefficient (AC) measurements. The study presents clear results on the accuracy of using a reduced number of acquisitions and the choice between mean and median values, which holds important clinical implications.

However, I feel there are some areas that require further clarification and expansion before the manuscript can be considered for publication.   Comments:

1. Demographic Data: Why are only females shown in the demographic table? It will be helpful if demographic details for male participants are also included  for clarity.

2. Acquisition Numbers: Why were only 5 and 3 acquisitions chosen for evaluation? Consider explaining the rationale behind selecting these specific numbers.

3. IQR/M Thresholds: Could you clarify why specific thresholds for IQR/M values (e.g., 15%) were chosen and how exceeding these affects diagnostic accuracy?

4. Technical Terminology: Provide brief explanations for technical terms like AUROC, IQR/M, and SD/A to make the paper accessible to a wider audience.

5. Formatting Issue: In the section listing performance criteria, "(j),(k)" is missing. Please correct the sequence for clarity.Sample Diversity: The study could benefit from a more diverse sample population. Consider addressing this in future studies.

6. Figures: Add brief interpretations to figures, particularly the statistical plots, for better understanding.

7. The study includes a relatively small number of patients with severe liver steatosis (S3), which limits the strength of the conclusions drawn for this subgroup. It would be beneficial to expand the discussion on how this limitation affects the study’s generalizability. Future studies should aim to include larger, more diverse populations to verify these findings across different demographic groups.

Author Response

Dear Reviewer,

General Comments: The manuscript titled "Assessing Quality of Ultrasound Attenuation Coefficient Results for Liver Fat Quantification" provides a valuable contribution to the field of liver fat quantification, offering a detailed evaluation of ultrasound-based attenuation coefficient (AC) measurements. The study presents clear results on the accuracy of using a reduced number of acquisitions and the choice between mean and median values, which holds important clinical implications.

However, I feel there are some areas that require further clarification and expansion before the manuscript can be considered for publication. 

R: Thank you for your evaluation.

Comments:

  1. Demographic Data: Why are only females shown in the demographic table? It will be helpful if demographic details for male participants are also included for clarity.

R: The number of females was higher than the number of males, and we followed the usual rule for reporting sex in tables in a scientific article. However, in the text of the revised manuscript (first sentence of the Results section), we have now reported the number of both females and males.

  1. Acquisition Numbers: Why were only 5 and 3 acquisitions chosen for evaluation? Consider explaining the rationale behind selecting these specific numbers.

R: In choosing the number of acquisitions for the analysis, we followed the recommendation of the WFUMB guidance on fat quantification (Ref.#5), i.e. three to five AC acquisitions.

  1. IQR/M Thresholds: Could you clarify why specific thresholds for IQR/M values (e.g., 15%) were chosen and how exceeding these affects diagnostic accuracy?

R: This is the threshold recommended by the WFUMB guidance for fat quantification (Ref. #5). The effect on accuracy of the different IQR/M thresholds, including IQR/M higher or lower than 15%, is reported in table 3 and table 4 of the submitted manuscript.

  1. Technical Terminology: Provide brief explanations for technical terms like AUROC, IQR/M, and SD/A to make the paper accessible to a wider audience.

R: For the AUROC, the abbreviation is spelled out the first time it is mentioned; this is a common statistic used to assess diagnostic accuracy, as stated in the M&M section of the submitted manuscript. Therefore, we do not believe it is necessary to provide a detailed explanation of this statistical method. In the Introduction section of the revised version, we have specified that IQR/M assesses the variability between consecutive acquisitions. SD/A assesses the variability between measurements when the average (mean) value is used. It is the coefficient of variation: this has been specified in the Introduction section of the revised manuscript.

  1. Formatting Issue: In the section listing performance criteria, "(j),(k)" is missing. Please correct the sequence for clarity. Sample Diversity: The study could benefit from a more diverse sample population. Consider addressing this in future studies.

R: Thanks for pointing out that: this mistake has been corrected. In the last sentence of the conclusion paragraph of the submitted manuscript we have underscored that to confirm these findings further studies in real-world settings are needed.

  1. Figures: Add brief interpretations to figures, particularly the statistical plots, for better understanding.

R: For each figure, the caption is displayed on the top. All the details regarding the interpretation of each one are given in the captions of the submitted manuscript.

  1. The study includes a relatively small number of patients with severe liver steatosis (S3), which limits the strength of the conclusions drawn for this subgroup. It would be beneficial to expand the discussion on how this limitation affects the study’s generalizability. Future studies should aim to include larger, more diverse populations to verify these findings across different demographic groups.

R: We have added this point to the limitations section: This series included a relatively small number of individuals with severe liver steatosis (S3), which may have limited the strength of the findings for this group.